

# Impact of coastal forcing and groundwater recharge on the growth of fresh groundwater resources in a mega-scale beach nourishment

Sebastian Huizer[1,2], Max Radermacher[3], Sierd de Vries[3], Gualbert H.P. Oude Essink[1,2], Marc F.P. Bierkens[1,2]

[1]Department of Physical Geography, Utrecht University, Utrecht, Netherlands
    [2]Department of Subsurface and Groundwater Systems, Deltares, Utrecht, Netherlands
    [3]Faculty of Civil Engineering and Geosciences, Department of Hydraulic Engineering, Delft University of Technology, Delft, Netherlands

*Correspondence to*: Sebastian Huizer (s.huizer@uu.nl)

**Abstract.** Large concentrated sand replenishments or nourishments are one of the few coastal protection measures that can simultaneously result in an increase of local fresh groundwater resources. For a large beach nourishment called the Sand Engine - constructed in 2011 at the Dutch coast - we have examined the impact of groundwater recharge and coastal forcing (i.e. natural processes that drive coastal hydro- and morphodynamics) on the growth of the fresh groundwater resources between 2011 and 2016. Measurements of the morphological change and the tidal dynamics were incorporated in a

calibrated three-dimensional and variable-density groundwater model of the study area. Simulations with this model showed that the detailed incorporation of the local hydro- and morphodynamics and the actual recharge rate can result in a reliable reconstruction of the growth in fresh groundwater resources. Similarly, the neglect of tidal dynamics, land-surface inundations and morphological changes in model simulations can result in considerable overestimations of the volume of fresh groundwater. In particular wave run-up and coinciding coastal erosion during storm surges limit the growth in fresh

groundwater resources in dynamic coastal environments, and should be considered at potential nourishment sites to delineate the area that is vulnerable to salinization.



## 1 Introduction

Groundwater is an important – in many situations vital – source of clean fresh water for most coastal communities in the world. However these coastal fresh groundwater resources are to an increasing degree affected by seawater intrusion, primarily caused by (excessive) groundwater extraction and sea-level rise (Ferguson and Gleeson, 2012; Taylor et al., 2013).

Global population growth in coming decades will lead to a rising demand for fresh water, and combined with the projected sea-level rise this will likely result in a gradual decline of fresh groundwater resources (Famiglietti, 2014; Wong et al., 2014). In addition, sea-level rise can also increase coastal flooding – caused by storm surges – and may lead to an increase in coastal erosion, which in turn will induce seawater intrusion and may cause a loss of wetland and biodiversity (FitzGerald et al., 2008; Passeri et al., 2015; Wong et al., 2014). Coastal lowlands with low topographic gradients and small islands are

particularly vulnerable, because these areas are the most susceptible to coastal flooding and seawater intrusion (FitzGerald et al., 2008; McGranahan et al., 2007; Michael et al., 2013; Rotzoll and Fletcher, 2012).

There are two potential responses to these rising threats to coastal communities, especially in relation to sea-level rise: (global) climate mitigation and (local) adaptation (Nicholls, 2011; Wong et al., 2014). With the progression of our

knowledge and expectations on sea-level rise, the international perspective has shifted to adaptation (Brown et al., 2014). Some countries, such as the Netherlands, Germany and the United States, have implemented coastal protection measures, which is the only adaptation approach that additionally might help to preserve fresh groundwater resources (van Koningsveld and Mulder, 2004; Rosenzweig and Solecki, 2010; Sterr, 2008).

In the Netherlands – a vulnerable low-lying country with a long history of coastal flood management – sandy shorelines have been successfully maintained and reinforced with an extensive sand nourishment program in the last decades (Giardino et al., 2011; Keijsers et al., 2015). The anticipation of sea-level rise has led to new adaptation measures (Kabat et al., 2009), where the construction of a large concentrated beach nourishment called the Sand Engine (also called Sand Motor) is one notable example (Fig. 1). This large beach nourishment was created on the Dutch coast as part of an effort to attain a more

sustainable coastal protection approach (Slobbe et al., 2013; Stive et al., 2013). Contrary to regular nourishment, the Sand Engine is deemed advantageous because it only causes a disturbance at a concentrated part of the coastline during a short time, after which the excess sand nourishes the larger length of the coastline gradually by natural along-shore sand transport. Apart from being ecologically advantageous, an important additional advantage of such large beach nourishments is that fresh groundwater resources may increase substantially during its lifespan (Huizer et al., 2016).


This study aims to evaluate the factors that determine the growth of fresh groundwater resources in large concentrated beach nourishments, through the reconstruction of the development of a fresh groundwater lens and mixing zone in the Sand Engine during the first five years of its existence. This reconstruction was conducted with a three dimensional (3D) variable-



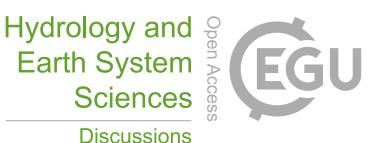

density groundwater model, and concentrated on the assessment of the impact of local conditions (e.g. tides and storm surges) on the spatial and temporal distribution of fresh groundwater. Considering the purpose of the project, we will focus in particular on the effects of coastal forcing (i.e. natural processes that drive coastal hydro- and morphodynamics such as wind, waves, and tides) and associated geomorphological changes of the nourishment.

In addition, this study analyses temporal changes in the quantity of fresh groundwater, through the evaluation of the contribution of prime factors (e.g. land-surface inundations, geomorphological changes, and groundwater recharge) to this volume of fresh groundwater. The paper first briefly describes the characteristics of the study area, and the adopted methodology for the analysis of the change in fresh groundwater resources. Next, the model calibration and model results are
described and examined, as well as the impact of land-surface inundations, morphology, and groundwater recharge on simulated fresh groundwater resources. Finally, the methodology and results are discussed, emphasizing the model uncertainties, and the general implications for the growth of fresh groundwater resources in large beach nourishments.

## 2 Data and Methods

### 2.1 Site description: Sand Engine

The Sand Engine ('Zandmotor' in Dutch) is a large concentrated beach nourishment of approximately 17 million m$^3$ sand, which was placed on the Dutch coast in 2011 as a hook-shaped peninsula (Fig. 1). This nourishment is part of an innovative pilot project in which this nourishment method is evaluated with respect to current nourishment methods in the Netherlands (i.e. large-scale distribution of sand). One appealing hallmark of the Sand Engine is that natural forces (i.e. wind, waves and currents) gradually transport the replenished sand along the retreating coast, and simultaneously support natural dune growth
(Slobbe et al., 2013).

Coastal forcing – storm surges in particular – led to substantial geomorphological changes at the Sand Engine in the measurement period. Between 2011 and 2016 the shoreline along the outer perimeter of the peninsula retreated approximately 200 m (de Schipper et al., 2016). The geomorphological changes of the Sand Engine were monitored every
one to three months with topographic surveys, as part of an intensive monitoring program. Spatial interpolations of all topographic surveys were used to update the surface elevation in the model, and these were implemented as sequential grid regenerations.

### 2.2 Variable-density groundwater flow model

Spatial and temporal changes in fresh groundwater in the Sand Engine were simulated with a 3D groundwater model, in
which the computer code SEAWAT was used to simulate variable-density saturated groundwater flow and salinity transport (Langevin et al., 2008). In SEAWAT the governing flow and solute transport equations are coupled and solved with a cell-





centred finite difference approximation. Numerous studies have applied this code to simulate variably-density, transient groundwater flow in coastal environments (Colombani et al., 2016; Holding and Allen, 2015; Pauw et al., 2014; Rasmussen et al., 2013; Webb and Howard, 2011).

The model domain had a length of 4500 m and width of 1500 m (Fig. 1), and was discretized into 75 rows and 225 columns with horizontal cell sizes of 20 m, and 28 layers with a thickness of 0.5 m in the upper layers and 1 m in layers below -7 m NAP (Amsterdam Ordnance Datum, which is approximately equal to MSL). Boreholes from the Sand Engine and adjacent dunes show that the subsoil of the study area consists of sandy aquifers with fine to coarse-grained sand, which are (partially) interrupted by two thin aquitards consisting of sandy clay, and are separated from underlying aquifers by an

aquitard consisting of clay and peat (Fig. 2). The replenished sand and upper aquifers (aquifer 1a and 1b) are mainly composed of medium coarse-grained sand, while the sand in the dunes and below -9 m NAP (aquifer 2) is mainly composed of fine-grained sand (Fig. 2). The underlying aquitard – situated between -17 and -20 m NAP – was defined as the local hydrogeological base of the model.

The boundaries of the model domain were defined either as a no-flow boundary (boundaries perpendicular to shoreline, and hydrogeological base) or as a specified head and concentration boundary (boundaries parallel to the shoreline). Specific head and concentration boundaries within the North Sea equalled tide gauge measurements in the harbours of Scheveningen and Hoek van Holland and seawater salinity of 28 g TDS L$^{-1}$ (i.e. equal to the observed average seawater salinity at the site: Rijkswaterstaat 2012). Other specified head and concentration boundaries were determined with an additional simulation

with the groundwater model as described in Huizer et al. (2016), where the groundwater recharge was adapted according to the model scenarios (Sect. 2.4.3). The groundwater head distribution at the start of the model simulations – before the completion of the Sand Engine in June 2011 – were set equal to the calibrated conditions of the same previously mentioned model (Huizer et al., 2016). The initial groundwater salinity distribution was approximated with a sharp (vertical) fresh-salt groundwater interface in the foredunes, because the former model underestimated the salinization close to the dunes and

because previous nourishments led to seawater intrusion in the (newly constructed) foredunes. The interface was positioned in the foredunes with fresh groundwater salinities of 0.1 g TDS L$^{-1}$ in the dunes, and completely saline groundwater in the foredunes, beach and Sand Engine (28 g TDS L$^{-1}$). Model cells close to this interface and in the dunes were excluded from evaluations of the growth of fresh groundwater resources in model scenarios.

In the adjacent dune area Solleveld a drinking water company extracts groundwater, and in order to prevent any undesirable or unexpected effects of the construction of the Sand Engine to the groundwater quality (e.g. flow of saline groundwater towards pumping wells), 29 pumping wells were installed in 2012 on the first dune ridge (see red points in Fig. 2). These pumping wells maintain the groundwater level at +0.8 to +1 m NAP with the aim to control the direction of the groundwater flow, and were included as such in the model simulations (Stuurman, 2010).





## 2.3 Model calibration

The groundwater model was calibrated with measurements of the groundwater head in monitoring wells 1 to 8 (Fig. 2). All these monitoring wells contain one well screen in the phreatic aquifer 1, and monitoring wells 2, 3, 7, and 8 contain a second well screen in underlying aquifers 2 (see cross-section in Fig. 2). The calibration was performed for measurements from 1

May 2014 until the end of the scenario simulations on 31 May 2016.

The calibration comprised of manual model parameter adjustments and comparisons of measured and simulated (transient) groundwater heads and groundwater salinities. For the evaluation of the fit to the measured groundwater heads, the subsequent calibration criteria were adopted: the error between the measured and simulated groundwater head should be

smaller than the observed variation in groundwater level (average standard deviation is 0.1 m in the calibration period), and the variation in the simulated groundwater head should be close or (almost) identical to the observed fluctuation pattern. For the calibration of the groundwater salinities similar criteria were adopted: the error between the measured and simulated groundwater salinity should be small or otherwise explicable, and the depth of the fresh to salt groundwater interface and mixing zone thickness should be close or (almost) identical. The calibration concentrated on a selection of model parameters:

hydraulic conductivity, storage coefficients, and dispersivity. These model parameters were adjusted with small incremental changes from an initial estimate, which was identical to a previous model calibration of the same area (Huizer et al., 2016). The dispersivity was adapted in accordance with the observed mixing zone thickness.

Ideally, we would have liked to split the data into a calibration and a validation dataset (split sample approach). However,

the number of observation locations and the length of the time series made such an approach unachievable, and therefore all available information was used for the calibration. This meant that only the lack of fit of the model could be verified, not the predictive uncertainty. The calibrated set of model parameters is shown in Table 1.

## 2.4 Model scenarios

Our hypothesis was that the growth of fresh groundwater resources (classified as 0-1 g TDS L$^{-1}$) in the Sand Engine, and

beach nourishments in general, is determined by six factors: land-surface inundations produced by tides and storm surges, geomorphological changes, groundwater recharge, geology (e.g. grain size, aquifer thickness), groundwater extractions, and groundwater inflow and outflow from or to the beach nourishment. In this study we limited ourselves to the evaluation of the spatial and temporal impact of the first three factors, with model simulations for the period of 1 June 2011 (completion Sand Engine) to 31 May 2016. All model scenarios consisted of small modifications of the calibrated model (i.e. reference case),

which are summarized in Table 2 and described in detail in Sect. 2.4.1 to 2.4.3.



### 2.4.1 Land-surface inundations

High frequency (10-minute time interval) tide gauge measurements in the harbours of Scheveningen and Hoek van Holland were used to estimate the seawater level near the Sand Engine (i.e. Still Water Level: SWL). The average seawater level of both measurement sites was used as an estimate of the local seawater level, and the tidal North Sea boundary was modelled

as 'General Head (head-dependent) Boundaries and Drains' (Mulligan et al., 2011). All model layers above -2 m NAP were defined convertible (saturated thickness) and rewettable with a wetting threshold of 0.05 m (McDonald et al., 1992). To ensure a reactivation of all inundated model cells, an additional seawater infiltration of 0.01 m per minute (equal to the vertical hydraulic conductivity) was added to the area of inundation during rising tides. In addition, to analyse the impact of tidal dynamics on fresh groundwater resources, a simulation (scenario A1) with a constant seawater level of 0.065 m NAP

(MSL simulation period) was executed (Table 2).

Wave set-up (i.e. local rise of the MSL) and wave run-up (i.e. maximum level of wave up-rush on the beach) will result in an increase in the extent of land-surface inundations, and hence to an increase of seawater intrusion. To assess the impact of wave set-up and wave run-up on fresh groundwater resources both processes were included in the reference model, and

excluded in model scenario A2 (Table 2). Wave set-up was modelled with an identical approach as the observed seawater level fluctuations, and wave run-up as an infiltration of seawater between the wave set-up height and the wave run-up height. The infiltration rate at the wave set-up height was estimated as the drainable storage (determined by specific yield) between SWL and the wave set-up height, and above the wave set-up height this infiltration rate was reduced linearly to a value of 10 % at the wave run-up height. The wave set-up and wave run-up height in every model period was estimated with the

parameterization for set-up on dissipative sites (Stockdon et al., 2006), which is dependent on the deep water significant wave height, and the deep water wave length. The deep water significant wave height and deep water wave length were estimated with offshore measurements, located 50 km southwest from the study site.

### 2.4.2 Geomorphology

The morphological evolution of the Sand Engine in the period 2011-2016 has led to significant decreases in the dimensions

of the tidal channel (hereafter referred to as lagoon), which gradually chocked the tidal system in the lagoon (de Vries et al., 2015). As a result the tidal amplitude decreased over time and the mean water level inside the lagoon increased (Fig. 3).

In an effort to produce a hindcast of tidal water levels inside the lagoon for the period of 3 August 2011 to 4 January 2016, the Delft3D flow model by de Vries et al. (2015) was extended (Fig. 4). Measurements of morphology and boundary

conditions (wind and water levels) were included in the simulation. Details on the model setup, assumptions and boundary conditions can be found in Appendix A. This hindcast of tidal water levels inside the lagoon was implemented in the reference model, where we assumed that the lagoon water level sustains over the whole channel. Before and after the



hindcasted period we assumed that the water levels inside the lagoon were identical to the offshore seawater levels. Note that after 4 January 2016 the chocking of the tidal system in the lagoon was (temporarily) lifted, because the continued erosion of the outer perimeter of the Sand Engine and the increasing hydraulic gradient between the lagoon and the North Sea led to breach of the sand barrier that separated the two systems.

To determine the impact of the observed morphological changes and the increase of the mean water level inside the lagoon on fresh groundwater resources, two additional model scenarios (B1 and B2) were implemented (Table 2). In model scenario B1 all morphological changes between 2011 and 2016 were excluded and hence the topography remained constant (equal to the situation in August 2011). In model scenario B2 the chocking of the tidal system in the lagoon was ignored and thus the

water level in the lagoon remained equal to the North Sea throughout the simulation period (model scenario B2).

### 2.4.3 Groundwater recharge

Hourly measurements of the precipitation at a measurement station in Hoek van Holland – located 9 km south-south-west from the measurement site – were used as an estimate of the precipitation on the Sand Engine. Potential soil evaporation was

calculated with the FAO Penman-Monteith method for hourly time steps, where the mean wind speed, air temperature, global radiation, and relative atmospheric humidity were also based on hourly measurements in Hoek van Holland (Allen et al., 1998). The FAO Penman-Monteith equation was adapted with estimations of the aerodynamic resistance and surface resistance for bare sand (Voortman et al., 2015).  Similar to Voortman et al. (2015) the ratio between the incoming solar radiation and the clear sky solar radiation between sunset and sunrise was linearly interpolated between the 4 to 6 h average

before sunset and after sunrise (Gubler et al., 2012). The resulting hourly precipitation and evaporation was linearly distributed over smaller-sized stress periods.

The actual soil evaporation was estimated with the average moisture content between surface elevation and the extinction depth, where the evaporation was set equal to the potential evaporation for moisture contents equal and larger to field

capacity. For moisture contents smaller than field capacity, the soil evaporation drops linearly to zero, parallel with the decrease in moisture. For coarse sand field capacity was estimated as 0.042 $cm^3/cm^3$ (Wösten et al., 2001).

The moisture content was calculated with a water budget method of precipitation and evaporation, where we assumed that percolation only occurs when the moisture contents equals field capacity. At that point the groundwater recharge equals the (positive) difference between precipitation and evaporation. Based on literature data the extinction depth was estimated as

0.5 m (Shah et al., 2007; Wösten et al., 2001). However, because of uncertainties in this estimation, we have also conducted model scenarios C1 and C2 with extinction depths of 0.25 and 0.75 m (±50%; Table 2).





The effects of sea spray deposition were estimated with semi-empirical equations (Stuyfzand, 2014) with wind speed and wind direction measurements in Hoek van Holland. For the angle of the coastal high water line we used the angle of the (straight) shoreline of 228°, as existed prior to the construction of the Sand Engine. Between June 2011 and May 2016 the resulting annual mean TDS concentration – caused by sea spray deposition – were respectively 0.11, 0.08, 0.06, and 0.04 g TDS $L^{-1}$ at 100, 200, 500, and 1000 m from the local mean high water (MHW) height of +1.09 m NAP. These TDS concentrations were linearly interpolated based on the distance from the MHW height on the Sand Engine, with a maximum concentration of 0.11 g TDS $L^{-1}$.

## 3 Results

The calibrated groundwater model (i.e. reference case) contained all elements of the previously described model scenarios that were deemed important for the growth of the fresh groundwater resources: estimates of wave set-up and wave run-up (Sect. 2.4.1), hindcast of tidal water levels inside the lagoon (Sect. 2.4.2), and a groundwater recharge that was based on an extinction depth of 0.5 m below surface (Sect. 2.4.3). The simulated (transient) groundwater head and groundwater salinity of this model were compared with measurements at the study site. In addition, the reliability of the adopted spatial discretisation was tested with a grid convergence test, consisting of simulations with lower and higher spatial resolutions. In the subsequent model scenarios the effects of coastal forcing, geomorphology, and groundwater recharge on the growth of fresh groundwater resources were examined with respect to this calibrated model.

### 3.1 Model evaluation

Figures 6 and 7 show that the simulated groundwater head closely resemble the observed fluctuation pattern at the eight monitoring wells (MW) on the Sand Engine. This demonstrates that the calibrated model is able to reproduce the observations with plausible model parameters (Table 1) and indicates that the groundwater dynamics are described satisfactorily. . Near the shoreline (MW 1, 2, 5, 7) and in the centre of the Sand Engine (MW 8) the similarity between the observed and simulated groundwater head is strongest, with RMS errors varying between 0.08 and 0.15 m. Closer to the dunes (MW 4 and 6) the RMS error increases slightly due to temporary underestimations in the simulated groundwater head. The only exception to this overall pattern is MW 3, which has a larger RMS error in comparison with the other monitoring wells. However, the simulated fluctuation of the groundwater head in MW 3 is similar to the measurements, and the larger RMS error is primarily a result of a systemic underestimation. This underestimation of the groundwater head at MW 3 is probably caused by mismatches in the local geology (e.g. finer sand, or variations in the position, thickness or conductivity of aquitards) that reduces the decline in groundwater head. This possibility is corroborated by the contrast in the observed and simulated groundwater head in the bottom aquifer at MW 2 and 3 (Fig. 7). The measurements indicate that in reality the attenuation of the tidal signal is stronger at MW 2 and weaker at MW 3, while MW 2 is situated closer to the shoreline than



MW 3. Thus, this suggests a stronger variability in the thickness, hydraulic conductivity or spatial distribution of aquitard(s) near these monitoring wells.

One of the likely causes of the temporary underestimations of the groundwater head in MW 4, 6 and 8 – and to a lesser extent MW 3 – is (temporary) deviations of the groundwater head at the inland model boundary, as for example between February and July 2015 and February and June 2016. In these periods the deviation is largest for MW 4, 6 and 8. The mismatches at the inland model boundary are most likely either caused by overestimations of the groundwater extractions in the dunes (i.e. changes in extraction rate throughout the year) or underestimations in the volume of groundwater recharge in the dunes.

Another notable deviation in the observed and simulated groundwater head is the rapid rise of the groundwater level in MW 1, 2, 3, 5, 6 and 7, during a storm surge on 22 October 2014 (see also Fig. 5). The rise of the groundwater level is smaller in the model simulations – especially in the monitoring wells that lie closest to the shoreline – and this is probably caused by an underestimation of the wave run-up height or seawater infiltration during this storm surge.

The simulated groundwater salinity was also compared with chloride measurements, taken at the monitoring wells (from soil samples taken during the construction of the wells) between 10 and 14 March 2014 (Fig. 8). For the conversion of the measured chloride concentrations to salinity, we have adopted the relation between chloride and TDS as found in the North Sea; 0.553 g Cl L$^{-1}$ in 1 g TDS L$^{-1}$ (Millero, 2003). The depth profiles of the groundwater salinity (Fig. 8) confirm the presence of a fresh groundwater lens on top of an otherwise saline aquifer. Only in MW 4 the measurements show a decrease in the groundwater salinity at depth, but it is uncertain whether this is the result of a single measurement error or an actual deviation from the overall observed pattern.

Figure 8 shows that the reference simulation closely resembles the observed groundwater salinity in most monitoring wells. However, the deviations suggest that the average North Sea salinity may be higher (MW 1, 2, 3, and 7), and indicate that the thickness of the fresh groundwater lens is slightly overestimated in the reference case. The cause of this slight overestimation will likely either lie in an overestimation of the volume of groundwater recharge or an underestimation of the salinization of fresh groundwater by coastal flooding (in particular during storm surges). The only exception is MW 6, where the depth of the fresh – salt groundwater interface is underestimated, and the likely cause for this deviation is a mismatch in the initial groundwater salinity near the dunes (e.g. overestimation of the salinization by nourishments).

**3.2 Grid convergence**

To test whether the adopted spatial model discretization returned reliable quantifications of the volume of fresh groundwater in the study site, we have conducted a grid convergence test. As described in Sect. 2.2, the reference model discretization consisted of a horizontal grid size of 20 m, and 28 layers with a variable thickness of 0.5 (upper layers) to 1 m (lower layers).



This discretization was tested with three additional simulations with higher and lower spatial resolutions: one with an increased vertical resolution of 0.25 to 0.5 m over 56 layers (S1), one with a decreased vertical resolution of 1 to 2 m over 14 layers (S2), and one with a coarser horizontal grid size of 30 m (S3). All the model parameters, initial conditions and boundary conditions of the simulations were equal to the calibrated model.

The simulations with the above mentioned spatial resolutions show similar increases of the volume of fresh groundwater during the simulation period (Fig. 9). Coarser spatial resolution (S2 and S3) resulted in lower volumes of fresh groundwater, and a finer vertical spatial resolution (S1) resulted in a nearly identical growth of the fresh groundwater volume. However, at the end of the simulation period (May 2016) the overall deviation in the simulated change in fresh groundwater is small (1.1-

1.16 million m$^3$). This convergence is probably mainly caused by aquitard 1 (Fig. 2), which hampers the growth of fresh groundwater resources in a large section of the study area. Thus, the additional simulations show that subsequent increases of the spatial resolution would lead to similar growth curves, which suggests that the model grid was sufficiently refined. The convergence with regard to the temporal model discretization was not tested, because stability constraints were used to calculate the length of transport timesteps.

**3.3 Scenario A: Land-surface inundations**

One of the processes that will inhibit the growth of fresh groundwater resources in the Sand Engine is coastal forcing, which is the driving force of land-surface inundations. Tides will lead to frequent land-surface inundations near the shoreline, and storm surges to occasional and more extensive inundations. In both cases the extent and duration of the inundations will depend on the intensity of the wind and wave forces, and local morphology. These land-surface inundations will limit the

growth of fresh groundwater resources, as illustrated in Figure 10 and 11.

Over the whole simulation period the incorporation of the tidal dynamics results in a substantial deviation in the volume of fresh groundwater (Figure 10). Frequent and occasional land-surface inundations – generated by tides and storm surges – lead to a repeated salinization of fresh groundwater in the intertidal and supratidal area. This pattern can also be observed in

the in- and exclusion of wave set-up and wave run-up in respectively the reference model and model scenario A2. The neglect of these processes leads to an underestimation of the extent of land-surface inundations and the infiltrated volume of seawater, especially during storm surges. This is illustrated with the changes of the areal extent of fresh groundwater resources and position of the maximum yearly wave set-up and wave run-up heights (contour lines) in Fig. 11. In addition, it is important to note that the model calibration suggests the wave run-up height or seawater infiltration during storm surges

was underestimated (Sect. 3.1).

Another important variable that reflects the divergence in the model scenarios is the groundwater table. The omission of tides and storm surges leads to an underestimation of seawater intrusion, and therefore an underestimation in the





groundwater levels. In model scenario A1 the groundwater level on 1 June 2016 is 0.4-0.6 m lower than the reference case - and observed groundwater levels. In turn, this leads to a reduction of submarine fresh groundwater discharge and a larger inflow of fresh groundwater from the adjacent dune area, which also contributes to the overestimation of the growth of fresh groundwater resources in the study area.

**3.4 Scenario B: Geomorphology**

Besides hydrodynamics, coastal forcing also drives morphodynamics. Geomorphological changes in the study area from June 2011 until May 2016 consisted of a substantial retreat of the shoreline along the outer perimeter of the Sand Engine (Fig. 1) and a gradual decline in surface elevations (Luijendijk et al., 2017; de Schipper et al., 2016). These morphological changes partially led to a direct loss of fresh groundwater due to coastal erosion, but to a greater extent to indirect losses

because of a shift and in some situations extension of the intertidal and supratidal area (Fig. 11). The simulation with a constant surface elevation and bathymetry (B1) shows that the absence of morphodynamics would have led to substantially higher fresh groundwater volumes in the study area (Fig. 12).

Another result of the morphological changes was the development of a lagoon, which led to a gradual decrease of the tidal

amplitude and increase of the mean water level in the lagoon from 2011 to 2016, as described in Sect. 2.4.2. The model simulations (reference case) show that this led to a small decrease in the growth of the fresh groundwater volume (70,000 m$^3$ on 1 June 2016) in comparison with model scenario B2 where the water level in the lagoon remained equal to the offshore sea-level (Fig. 12).

The effect of the morphological evolution of the lagoon on the overall growth in the volume of fresh groundwater is relatively small in comparison with model scenario A (Sect. 3.3) and C (Sect. 3.5), because the inundation extent during storm surges (e.g. high seawater levels) is similar in both situations. The smaller growth of the fresh groundwater lens is primarily caused by a rise of the local groundwater level (around the lagoon), which led to a reduced inflow of fresh groundwater from the adjacent dune area and an increase in (lateral) seawater intrusion.

**3.5 Scenario C: Groundwater recharge**

On average the yearly precipitation from June 2011 until May 2016 was 938 mm, of which 421 mm fell in March-August (spring-summer) and 517 mm in September-February (autumn-winter). The average yearly potential soil evaporation was 990 mm, of which 695 mm in March-August and 295 mm in September–February. Thus, the net surplus based on the potential soil evaporation was -52 mm per year, and the actual evaporation rate will be important (especially in spring and

summer) for the net groundwater recharge and consequently the change in fresh groundwater resources in the study area.





Based on the available soil moisture between the surface and the extinction depth (see Sect. 2.4.3), the actual evaporation rate was determined. The resulting groundwater recharge for the respective extinction depths of 0.25, 0.5 and 0.75 m, as shown in Fig. 13, varied between respectively 595 mm (+78 mm relative to the reference case), 516 mm, and 470 (-46 mm relative to the reference case) mm per year. Most of the groundwater recharge (30 to 34%) occurred in autumn and winter

seasons, and least (9-12 %) in spring. Therefore, as to be expected, larger extinction depths led to more evaporation and less groundwater recharge, in particular in the spring and summer seasons.

The simulated change in the fresh groundwater volume (Fig. 14) shows that groundwater recharge is one of the primary driving mechanisms, with increases in periods with relatively high percolation rates and a stabilisation or decrease in periods

with relatively low percolation rates. Parallel to groundwater recharge, most of the overall growth in the fresh groundwater lens occurs in the autumn and winter seasons. In periods with low recharge rates, the loss of fresh groundwater – primarily due to submarine groundwater discharge and coastal erosion – leads to a stabilisation or decrease in the overall fresh groundwater volume in the study area. One notable example is the change in the fresh groundwater volume over the period March 2014 to November 2015, which coincides with a substantially lower than average groundwater recharge (Fig. 13).

**4 Discussion**

The model simulations show that the growth of fresh groundwater resources in the Sand Engine was affected by both coastal forcing and groundwater recharge. Periods with high rainfall (mainly in autumn and winter) led to sharp increases in the fresh groundwater volume, while periods with little or no rainfall (mainly in spring and summer) led to a net loss of fresh groundwater. Coastal forcing led to land-surface inundations and considerable geomorphological changes in the study area,

where inundation events resulted in the salinization of the intertidal and supratidal areas and morphodynamics led to a shift or in some instances extension of these areas. Storm surges in particular are important for the determination of the fresh groundwater resources, because these are the primary drivers of coastal erosion and result in the most extensive land-surface inundations.

Comparisons with simulations that excluded coastal hydro- and morphodynamics showed that the incorporation of these processes was essential for a good calibration result and a reliable estimate of the intertidal and supratidal area, and thus essential for the estimation of the growth of the fresh groundwater resources. Besides land-surface inundations, tidal dynamics play an important role in the height and variability of the groundwater head. The exclusion of these tidal dynamics in the simulation led to substantial underestimation of the groundwater heads (between -0.4 and -0.6 m). Where the

simulation with a constant sea-level (equal to MSL) indicate a continued growth in fresh groundwater resources, the model including tidal dynamics and wave setup only indicates a clear (net) growth in fresh groundwater resources between June 2011 and December 2013, and a slight (net) growth or stabilisation in the period thereafter.





For most monitoring wells the simulated transient groundwater head and groundwater salinity corresponded well with measurements. This demonstrates that the hydro- en morphodynamic conditions of the study area could be reproduced with the adopted methodology. Most of the discrepancies between measurements and simulations could be explained by unmapped geological heterogeneity (near MW 3), mismatches in the initial groundwater salinity distribution (near the coastal dunes), and temporal variations in North Sea salinity (varies due to freshwater influence of the nearby river Rhine). It is important to note that the simulated groundwater salinity could only be compared with groundwater salinity measurements that were conducted in March 2014. Therefore it is uncertain to what extent the simulated change in groundwater salinity corresponds with the salinity distributions of the period thereafter.

While the close similarity between the measured and simulated groundwater head time series under realistic hydrogeological parameter settings indicate that the tidal dynamics and extent of land-surface inundations are well represented in the model, the modelled wave set-up height and wave run-up height (and related infiltration of seawater) probably differed strongly with reality at times. Small variations in surface elevation, bathymetry, wave height, and wave period could have led to substantial variations in wave set-up and wave run-up. Underestimations of the increase of the groundwater level during some of the storm surges (see Fig. 6) also indicate an underestimation of wave set-up height, wave run-up height or the infiltration of seawater. However, these differences could also be caused by deviations in the modelled and actual morphology at the study site. Despite the frequent topographical measurements, interim morphological changes were not observed. This especially affects periods with rapid and extensive morphological changes, e.g. periods with frequent or intensive storm surges. However, based on the calibration results and comparisons with on-site observations we believe that the adopted approach attained realistic but rough estimates of the effects of wave set-up and wave run-up on the coastal aquifer.

Being the main source of fresh groundwater, groundwater recharge is another important control on the development of the fresh groundwater resources of the sand engine. Vegetation was virtually absent on the Sand Engine, with the exception of a few pockets of dune grass in the last years of the simulated period, and therefore the processes that determine the groundwater recharge rates could be limited to precipitation and soil evaporation. Because the depth to the groundwater table was relatively large (larger than 1.5 m below surface in areas that were never inundated) in relation to the expected extinction depth, capillary rise will be limited or non-existent and was therefore neglected. Given these simplifications and the aim of this paper, we opted to estimate recharge rates with the described water budget method and analyse the uncertainty with the alterations to the extinction depth.

For future studies on the (potential) growth of fresh groundwater resources in coastal areas, it is recommended to monitor wave set-up, wave run-up, and seawater infiltration and evaluate the accuracy of the adopted approach in this research under





various conditions. However, this will require frequent topographic measurements to monitor morphological changes in the coastal area, e.g. one to three monthly topographic surveys as were executed on the Sand Engine. In addition, simulations with unsaturated groundwater models could provide more detailed estimates of the growth in fresh groundwater resources in these areas, because of a potential improvement in the simulation of groundwater recharge and seawater infiltration.

## 5 Conclusions

Between 2011 and 2016 the growth of the fresh groundwater lens in the study area was primarily determined by the groundwater recharge, (maximum) land-surface inundations due to storm surges, groundwater in- and outflow and to a lesser extent by geomorphological changes.

1. Groundwater recharge was the primary contributor to the growth of fresh groundwater in the large concentrated beach nourishment. An accurate estimation of the actual soil evaporation – besides meteorological measurements – will likely be important for the determination of the net input of fresh water in any coastal area.

2. Storm surges produced the most extensive land-surface inundations, and the coinciding infiltration of seawater resulted in a salinization of most of the fresh groundwater volume within this inundation area. The model simulations showed that (accurate) estimates of the maximum wave set-up and wave run-up height are important to delineate the area that is vulnerable to seawater intrusion and reconstruct the growth of fresh groundwater resources.

3. The groundwater level, or better, the hydraulic gradients within the study area determined the inflow of fresh groundwater from adjacent dunes and outflow of (fresh) groundwater via submarine groundwater discharge. Model simulations that underestimate the height of the groundwater level, e.g. by the neglect of tidal dynamics, can therefore lead to considerable overestimation of the fresh groundwater volume.

4. Finally, the geomorphological changes led to a gradual decline of the area that was not affected by seawater intrusion. Together with the increase of the mean water level inside the lagoon, this led to a small restriction of the potential growth in fresh groundwater resources.

In conclusion, the incorporation of hydro- and morphodynamics and accurate estimation of groundwater recharge rate are essential for a reliable estimate of the growth of fresh groundwater resources in dynamic coastal environments.

**Appendix A: Model setup Delft3D**

The depth-averaged numerical model was constructed with the modelling package Delft3D (Lesser et al., 2004), which numerically integrates the shallow water equations. Prescribed water levels along the off-shore boundary were obtained from permanent tidal stations south (Hoek van Holland) and north (Scheveningen) of the Sand Engine. Lateral boundaries were



forced with Neumann conditions, which represent the alongshore water level gradient. Bottom friction is specified with a spatially uniform Chezy coefficient of 65 $m^{1/2}\,s^{-1}$. A constant eddy viscosity of 0.01 $m^2\,s^{-1}$ was applied. The influence of wind was taken into account as a wind shear stress at the free surface.

The model bathymetry and associated nourishment geometry was constructed from a set of 34 bathymetric field surveys at the Sand Engine and the adjacent coastal cell. These surveys were conducted using a jetski-mounted single-beam echo sounder for the submerged part of the domain and a real-time kinematic differential GPS mounted on an all-terrain vehicle for the dry beach (de Schipper et al., 2016).

**Acknowledgements and Data**

This research is supported by the Dutch Technology Foundation STW, which is part of the Netherlands Organization for Scientific Research (NWO), and which is partly funded by the Ministry of Economic Affairs. This work was carried out within the Nature-driven nourishment of coastal systems (NatureCoast) program. The data used to produce the results of this paper may be obtained by contacting the corresponding author.

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



**Table 1.** Calibrated parameter values implemented in the model simulations.

| Layer type | Parameter | Sand Engine | Dunes |
|---|---|---|---|
| All model layers | Longitudinal dispersivity | 0.1 m | |
| | Transverse dispersivity | 0.01 m | |
| | Effective porosity | 0.30 | |
| | Specific storage | $2 \cdot 10^{-4}$ | |
| Phreatic aquifers | Horizontal hydraulic conductivity | 28.8 m d$^{-1}$ | 10 m d$^{-1}$ |
| 1a: [above -6 m NAP] | Vertical hydraulic conductivity | 14.4 m d$^{-1}$ | 5 m d$^{-1}$ |
| 1b: [-6.5 to -8 m NAP] | Specific yield | 0.20 | |
| Aquitards | Horizontal hydraulic conductivity | $5.76 \cdot 10^{-2}$ m d$^{-1}$ | |
| 1: [-6 to -6.5 m NAP] | Vertical hydraulic conductivity | $5.76 \cdot 10^{-3}$ m d$^{-1}$ | |
| 2: [-8 to -9 m NAP] | | | |
| Aquifer | Horizontal hydraulic conductivity | 10 m d$^{-1}$ | |
| 2: [-9 to -17 m NAP] | Vertical hydraulic conductivity | 5 m d$^{-1}$ | |

**Table 2.** Summary of model scenarios (grey items are equal to reference case)

| Model scenario | North sea water level | Lagoon water level | Extinction depth |
|---|---|---|---|
| Reference | Tide gauge + wave run-up | Hindcast Delft3D model | 0.5 m-surface |
| A : Inundation (Sect. 2.4.1) | 1 : MSL 2011-2016 (constant sea-level) 2 : Tide gauge (no wave set-up or run-up) | Hindcast Delft3D model | 0.5 m-surface |
| B : Morphology (Sect. 2.4.2) | 1 : Constant morphology / no morphological change Tide gauge + wave run-up | 2 : Tide gauge (equal to North Sea) | 0.5 m-surface |
| C : Recharge (Sect. 2.4.3) | Tide gauge + wave run-up | Hindcast Delft3D model | 1 : 0.25 m-surface 2 : 0.75 m-surface |

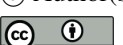



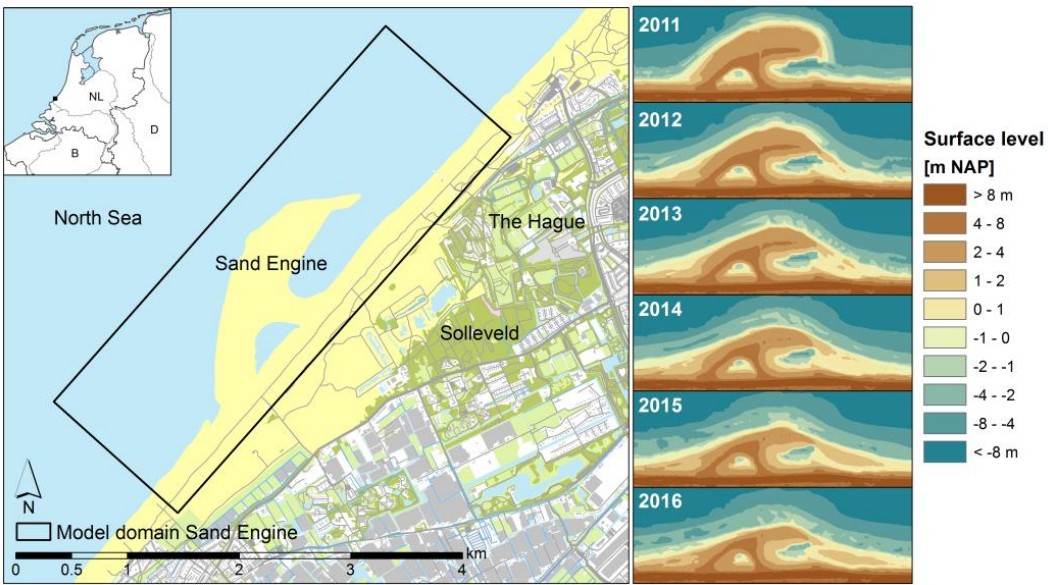

**Figure 1.** Map of the study area with the model domain (left), and the change of the surface level in the study area (morphological development) between 2011 and 2016 (right) in meters with respect to the NAP datum, which is approximately equal to mean sea-level.

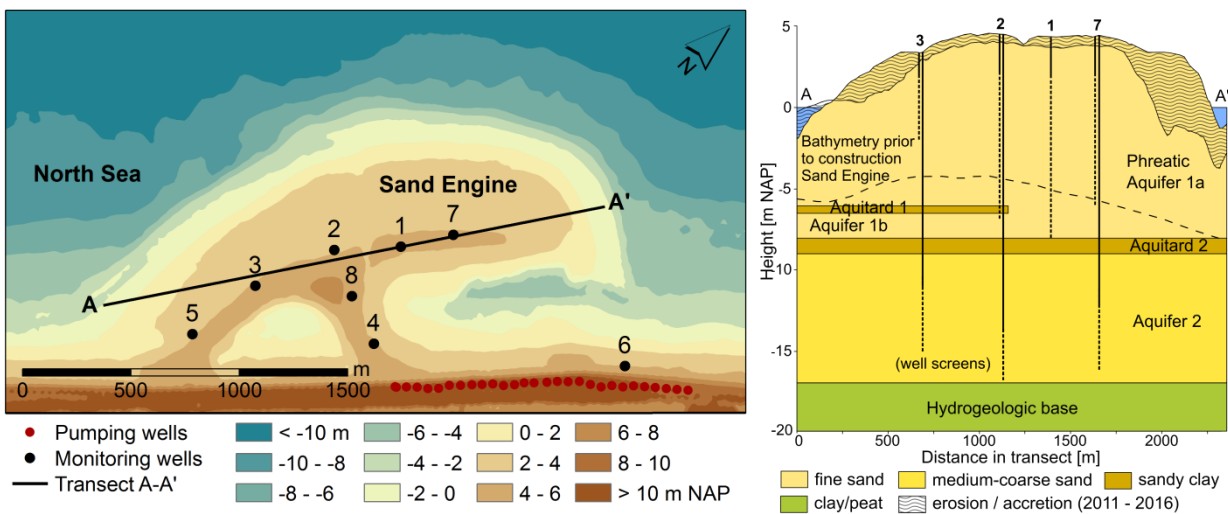

5   **Figure 2.** Contour map of topographic height (in meters with respect to the NAP datum) and hydrogeological cross section (along black line, A – A') of the Sand Engine between 1 and 3 August 2011, with the location of monitoring wells (black points) and pumping wells (red points). The dashed line in the cross section marks the bathymetry prior to the construction of the Sand Engine, and the wave-shading pattern marks the erosion or accretion of sand between 3 August 2011 and 3 June 2016.





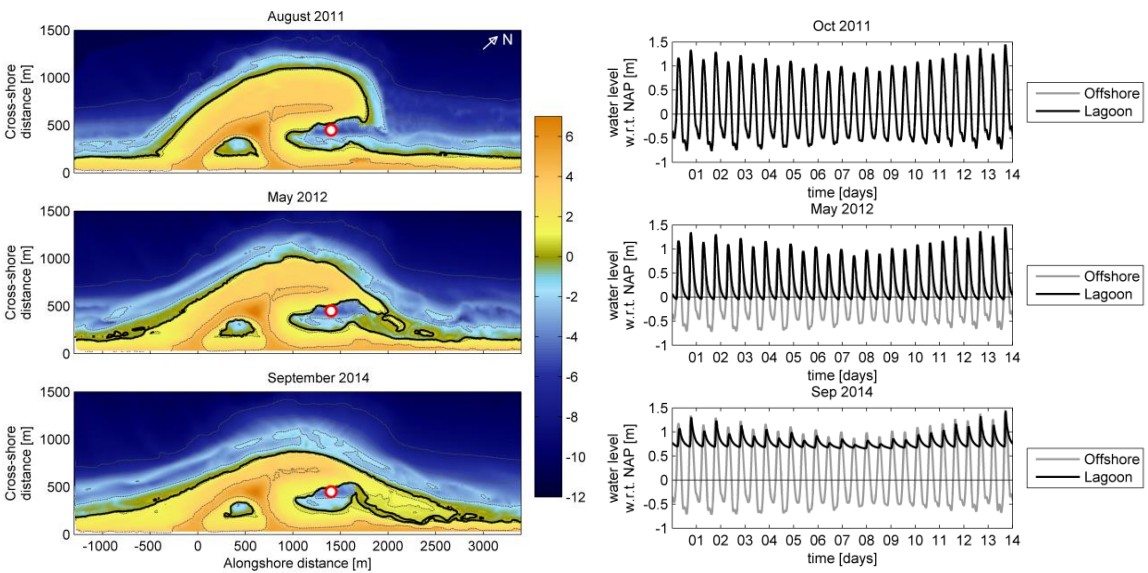

**Figure 3.** The images show three measurements of distinct morphologies of the Sand Engine domain in 2011, 2012 and 2014. The red circle represents the location where tides inside the lagoon are calculated with the Delft3D flow model in de Vries et al. (2015). The graphs show the simulated tides inside and outside the lagoon as obtained with the three successive morphologies.

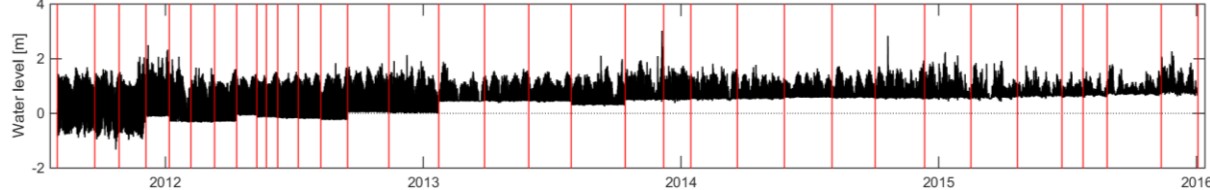

**Figure 4.** Simulated water level (in m NAP) inside the lagoon from 3 August 2011 to 4 January 2016, where the vertical red lines signify topographic surveys.

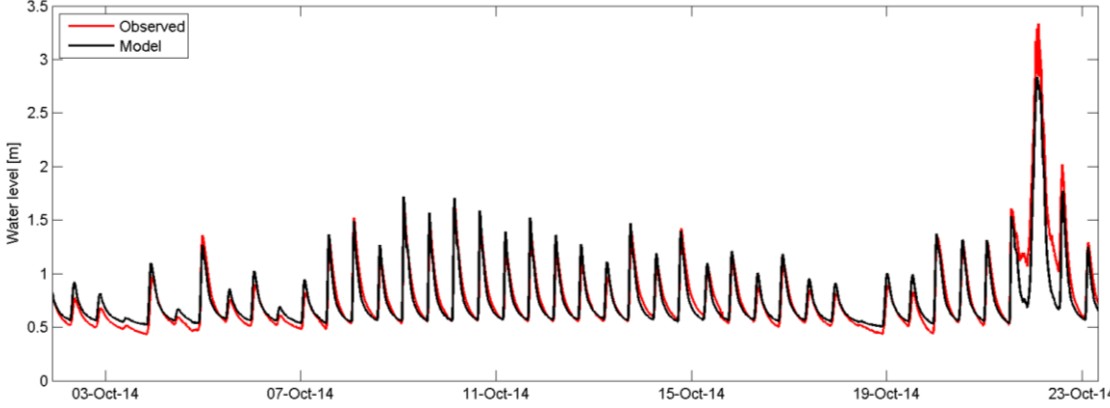

**Figure 5.** Observed and simulated water levels in the lagoon from 2-23 October 2014, which contains a fortnightly spring-neap tidal cycle and a storm surge around 22 October.





**Figure 6.** Observed and simulated groundwater level in MW 1–7 (phreatic aquifer 1: see Figure 2), from May 2014 to June 2016.





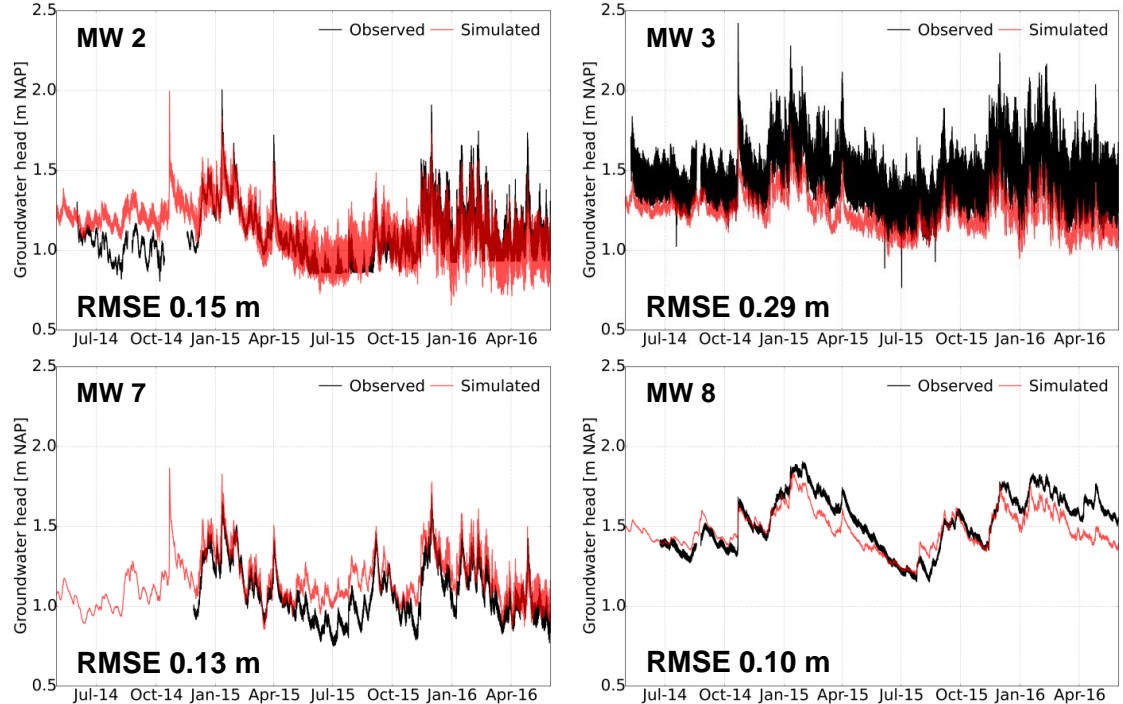

**Figure 7.** Observed and simulated groundwater head in MW 2, 3, 7 and 8 (in aquifer 2: see **Error! Reference source not found.**), from May 2014 to June 2016.





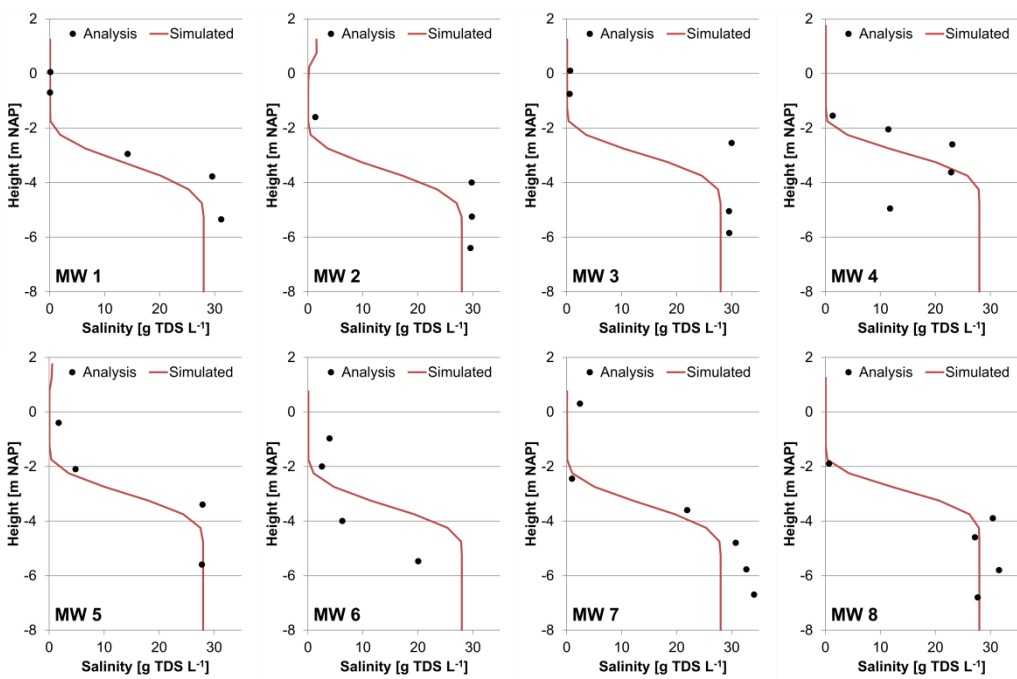

**Figure 8.** Depth profiles with the observed (black points) and simulated (red line) groundwater salinity in MW 1-8, obtained between 10 and 14 March 2014.

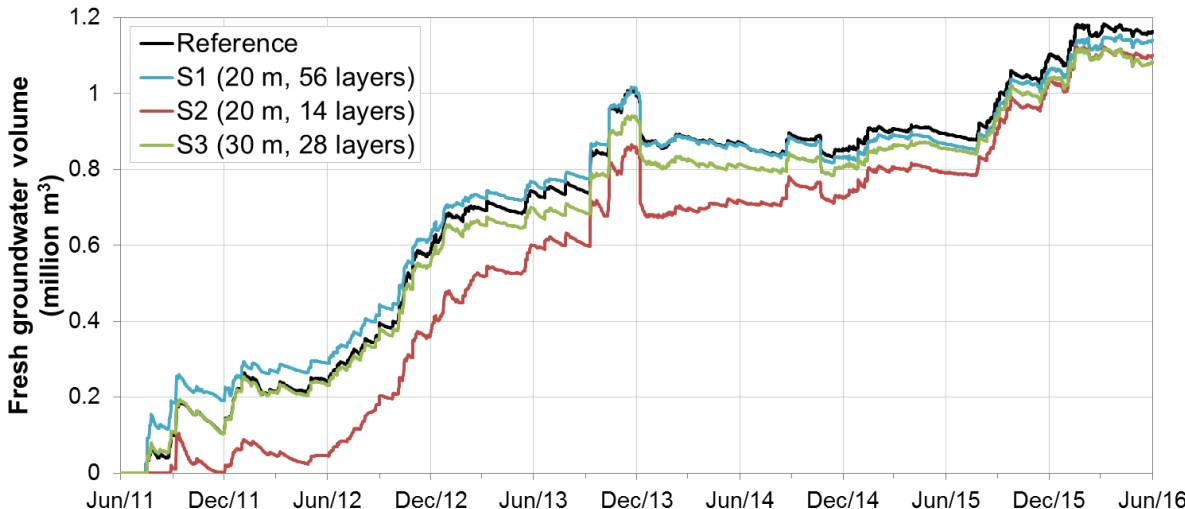

5   **Figure 9.** Change in the volume of fresh groundwater from June 2011 to May 2016 for the calibrated model (reference case) and model convergence simulations S1, S2 and S3.



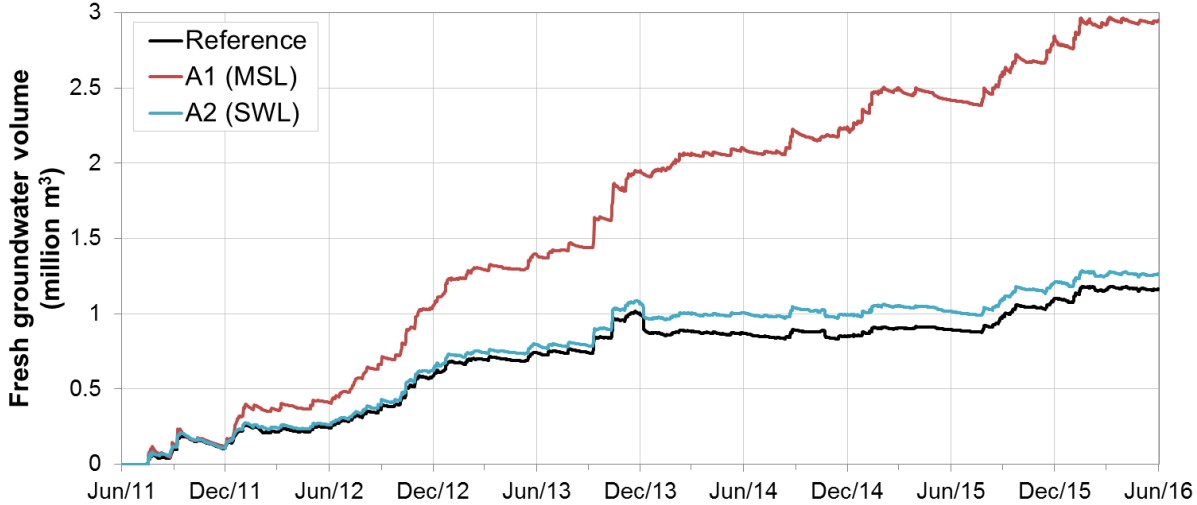

**Figure 10.** Simulated increase in the volume of fresh groundwater in the model domain from June 2011 to May 2016, for the reference model (incl. wave set-up and wave run-up), model scenario A1 (constant MSL), and model scenario A2 (SWL).

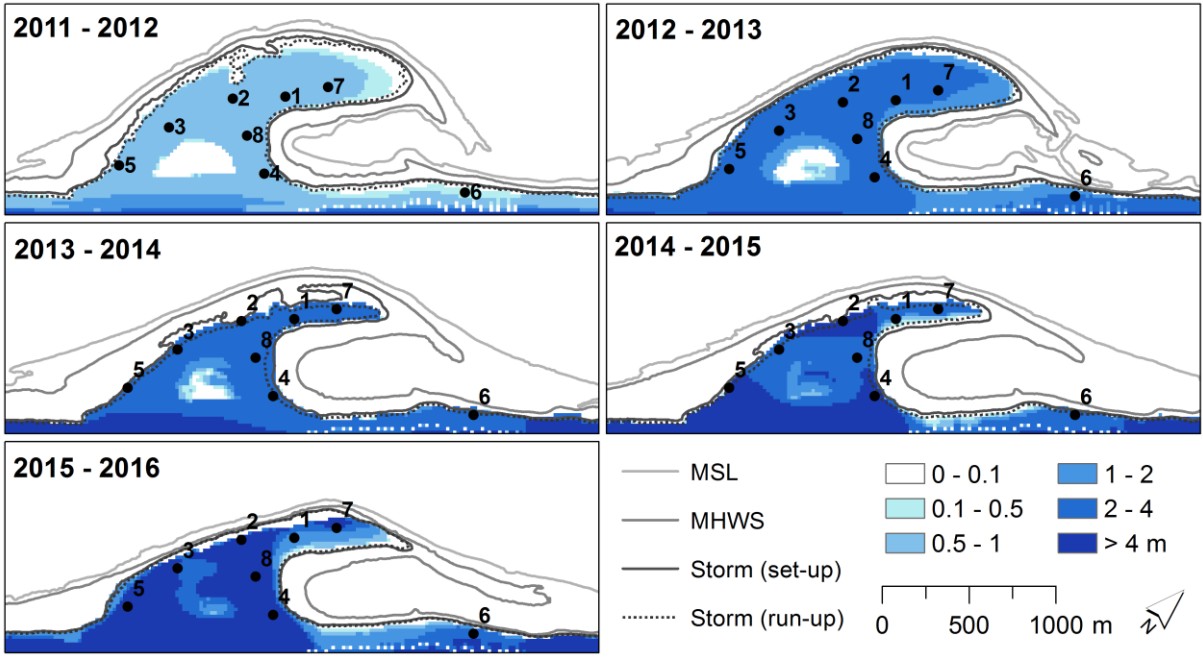

**Figure 11.** Thickness of fresh groundwater lens [m] 1, 2, 3, 4 and 5 years after the construction of the Sand Engine, with contour lines of the MSL, MHWS, and (estimated) maximum wave set-up and wave run-up height of every yearly period.




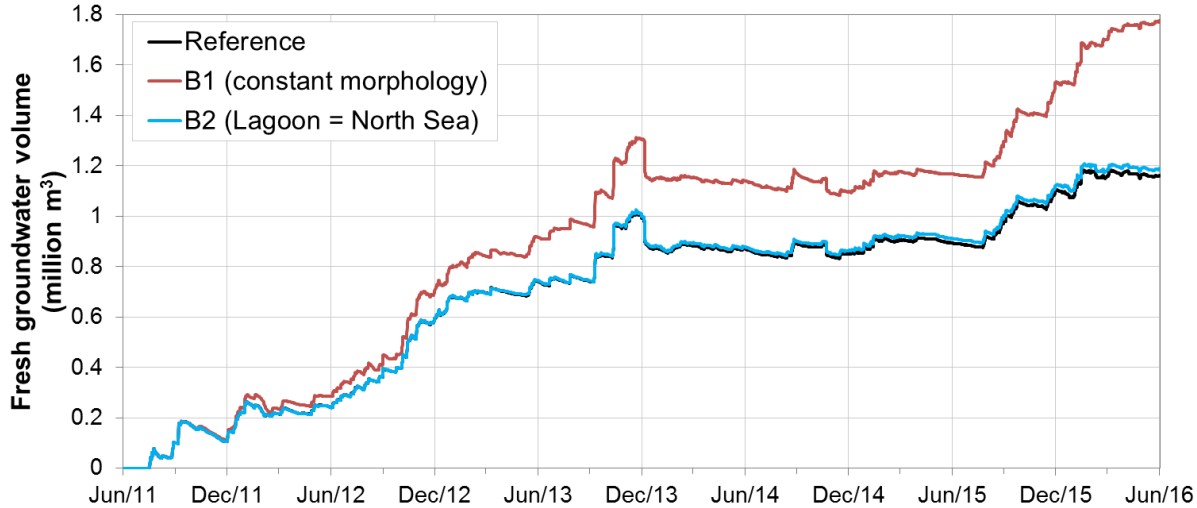

**Figure 12.** Increase in the volume of fresh groundwater from June 2011 to May 2016, for the reference case (with hindcast of the lagoon water level), model scenario B1 (constant morphology), and model scenario B2 (lagoon water level equal to North Sea).

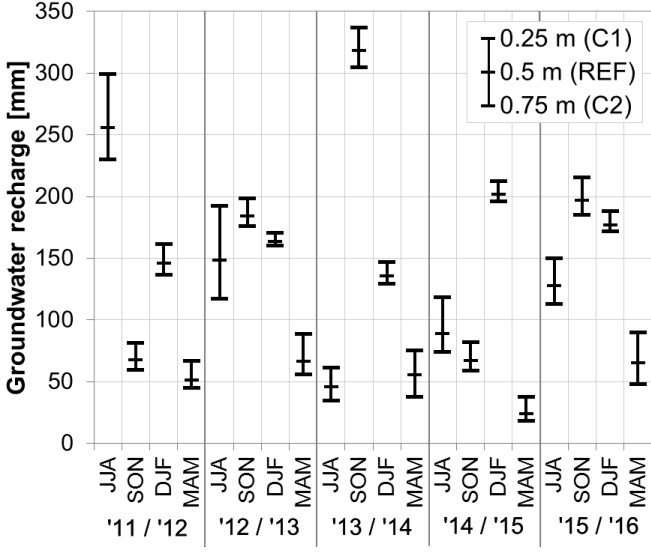

5 **Figure 13.** Simulated groundwater recharge per season (JJA: summer, SON: autumn, DJF: winter, MAM: spring) from June 2011 until May 2016, for the reference model (extinction depth 0.5 m), model scenario C1 (extinction depth 0.25 m: less evaporation), and model scenario C2 (extinction depth 0.75 m: more evaporation).

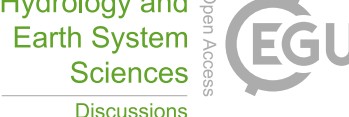

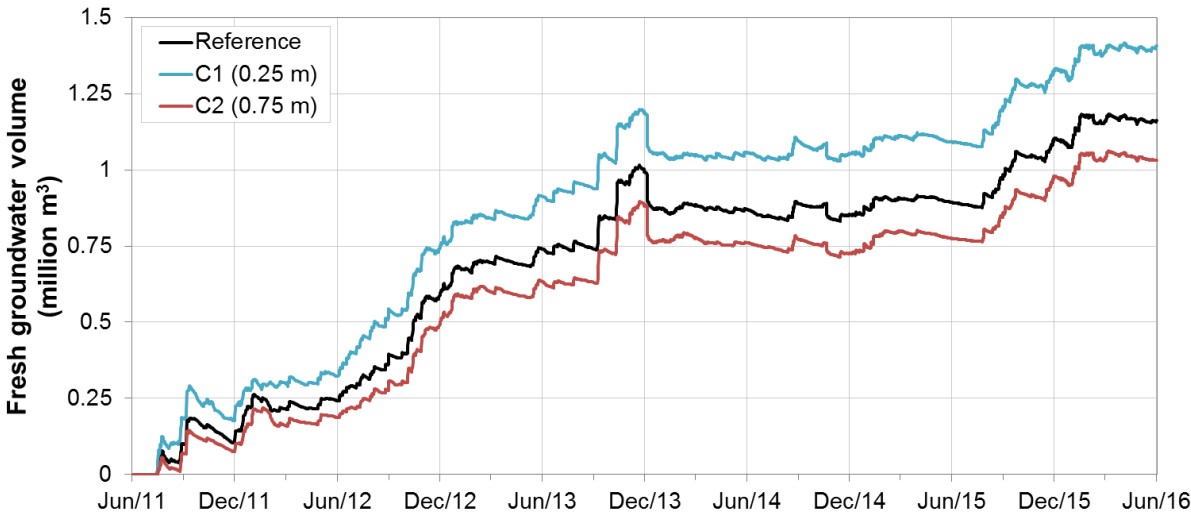

**Figure 14.** Increase in the fresh groundwater volume from June 2011 until May 2016, with the calibrated model with an extinction depth of 0.5 m (Sect. 2.4.3), a 50 % decreased extinction depth of 0.25 m (C1), and a 50 % increased extinction depth of 0.75 m (C2).