# Peer review of "Impact of coastal forcing and groundwater recharge on the growth of a fresh groundwater lens in a mega-scale beach nourishment"

_Hydrology and Earth System Sciences, 2017_

## Referee Comment (RC1) · Anonymous Referee #1 · 17 Sep 2017

This manuscript deals with a relatively young but interesting field of expertise: the hydrology of large beach nourishments. There is limited scientific research reported to date dealing with this topic, apart from a few earlier papers by the these authors (HESS, 2016; WRR, 2017). This manuscpript presents interesting (modelling) insights on the development of freshwater bodies below these beach nourishments. The content is scientifically relevant, of good quality, and properly presented. Some minor revisions are, however, required.

1. Being the third modelling paper on this topic in a row. The reader would be pleased with a clear picture of what is presented in which paper and what the differences between the applied groundwater models are. The WRR paper isn't even mentioned in the current manuscript.

2. The reader would be pleased with more quantitative information, mainly in Section 3, especially on the freshwater volume developed. For instance: not 'substantially higher fresh groundwater volumes' (P11, line 11/12), but a percentage of volume increase. Another example: P10, line 20 and 21.

3. Parts of Section 3 would better fit in Section 2.

Aspects to consider (HESS):

1.Does the paper address relevant scientific questions within the scope of HESS? Yes

2.Does the paper present novel concepts, ideas, tools, or data? Yes, but readers might doubt this because of the previous 2 papers. This should be better clarified by the authors.

3.Are substantial conclusions reached? Yes

4.Are the scientific methods and assumptions valid and clearly outlined? Yes

5.Are the results sufficient to support the interpretations and conclusions? Yes

6.Is the description of experiments and calculations sufficiently complete and precise to allow their reproduction by fellow scientists (traceability of results)? Yes

7.Do the authors give proper credit to related work and clearly indicate their own new/original contribution? No, see #2

8.Does the title clearly reflect the contents of the paper? Yes

9.Does the abstract provide a concise and complete summary? Yes, but also here: more quantication is required.

10.Is the overall presentation well structured and clear? No, part of the methods are found in Section 3

11.Is the language fluent and precise? OK. But excessive use of brackets. Are they really needed???

12.Are mathematical formulae, symbols, abbreviations, and units correctly defined and used? OK

13.Should any parts of the paper (text, formulae, figures, tables) be clarified, reduced, combined, or eliminated? No

14.Are the number and quality of references appropriate? No, one relevant paper is lacking, even though it is written by the same authors...

15.Is the amount and quality of supplementary material appropriate? Yes

Minor comments: - Title (and rest of manuscript): is 'resources' really the best term? I would associate that with use (abstraction) of the freshwater, which is not the case. Perhaps 'lenses' or 'bodies' are better alternatives - Intro: first sentence can be removed - P1, line 17: rmove 'similarly,' - P2, line 2, replace 'clean' by 'high-quality' - P2, line 14: do you really mitigate 'climate'? Suggest to replace 'climate' by 'climate change'. - P3, line 6. remove 'this study analyses' and add 'are analysed in this study' after 'fresh groundwater' - P3, line 7: remove e.g. and state only what you have analysed - P3, line 8: 'In this paper are described' instead of 'This paper describes' - P3, line 26: add 'groundwater' before 'model' - P3, line 29: add 'and salt' after 'fresh' - P4, line 10: use 'Aquifer 1a' instead of 'aquifer 1a'. Some holds for 'Monitoring well'. Capitals - P4, line 24-28: Vague, please add a figure to support this section - P4, 33: replace 'maintain' by 'keep' - P4, line 34: why is this control of groundwater direction needed? - P6, line 16: where was the wave set-up and run-up based on? - P6, line 18-22: make schematization and give max, min, average. - P6, line 28: no water level measurements available in the lagoon? - P7, line 21: 'smaller-sized' –> quantify - P8, line 21: sentence missing? - P9, line 8: did you check of these changes in abstraction rates might have occured? - P9, line 15-21: This part more or less just drops in here. Partly methods... - P9, line 28-29: Please provide more information on this initial

groundwater salinity (figure) - P10, line 1-4: consider to move this to methods section - P10, line 21-30: separate field observation and models results. Now, this part is confusing. Start with field observations. - P11, line 21: A1 or A2? C1 or C2? - P11, line 29: replace 'will be' be 'was' - P12, line 29: quantify - P13, line 6: North Sea salinity: speculative, you don't know if this can really explain deviations - P13, line 9: Why aren't there more measurements? - P13, line 31: any reference to studies with comperable approach? - P14, line 2: what is the basis for 'one to three monthly' - P14, line 3: what is a 'unsaturated groundwater model'? - P14, line 6: 'the study area' by 'a mega-scale beach nourishment (the Sand Motor)' - P14, line 10: Please introduce these bullets with one sentence.

---

## Referee Comment (RC2) · Anonymous Referee #2 · 24 Sep 2017

This manuscript shows the application of numerical models to simulate different scenarios for the quantification of the freshwater resources in a mega beach nourishment experience in the Dutch coast. This experiment, also called the Sand engine, has different challenges for the simulation of the groundwater flow since it is a highly dynamic environment that affects to the shape and boundary conditions, this is an interesting topic in the hydrogeological sciences and many readers can be interested on seeing the results. It is well written and applies the correct methods. But I think that the manuscript can be improved by showing the results with a more global perspective and less as a case study description. Also the novelty of this study should be presented more clearly and the analysis of the results/discussion can be improved.

The aims are very broad and the particular effects of coastal forcing and the geomorphological changes can be better described. I think that introducing the challenges from the perspective of the development of the numerical model would be needed to have a more complete overview. A more extensive introduction would help to understand the tasks that are going to be solved later.

In general it seems to be repetitive and a bit ambiguous the description of coastal forcing with multiple mentions of how important is this instead of describing precisely to what aspects is refereeing.

The boundary conditions of the model are based on previous models, unless this model is just a modification of a previous model (in that case it should be said), the boundary conditions should be clearly established. For example: "Other specified heads and concentration boundaries were determined with an additional simulation with the groundwater model as described in Huizer (2016)" Which are the other specified heads and concentrations? What model scenarios are referring here? (the scenarios are presented later in the manuscript but at this point is not clear) ". . . the former model underestimated the salinization. . ." which model? In general this paragraph can be rewritten under the consideration that the reader does not have to be necessarily familiar with the previous models in the area. Probably a full description of previous models and the novelties of this study would help to the reader to frame better this study.

It is no clear what are the areas presented as foredunes, dunes, beach and sand engine, they should be defined and presented in the map.

The model calibration presents some very clear criteria combined with ambiguous and arbitrary i.e."..the variation in the simulated head should be close or (almost) identical to the observed fluctuation pattern." What is close or identical? This is a very arbitrary description that can be improved. Another element that is arbitrary in the calibration is: "the salinity should be small or otherwise explicable". What is small? What is explicable?

It is defined that six factors affect to fresh groundwater resources but only three are considered. It is not clear what is the criteria for this and if this would also affect to the results obtained. I think that a better introduction would help to understand this.

The discussion is too descriptive basically presenting the results of the different models and adding some elements that could affect to the models (and in most cases are minor). A probably more interesting discussion, that can be also included in the conclusions, would be a quantitative comparison between the different factors that have been presented in this work. This would generate a broader impact of the results.

Minor comments

Page 3. Lines 5-12. The description about the outline of the work is not needed. Page 6. Lines 20-22. Which data were used for this calculation? Page 8. Line 21. There are two dots in a row. Page 9. Line 32. Either mention as described in sect 2.2., or repeat the model discretization but both are repetitive.

---

## Author Comment (AC1) · 9 Nov 2017

"This manuscript deals with a relatively young but interesting field of expertise: the hydrology of large beach nourishments. There is limited scientific research reported to date dealing with this topic, apart from a few earlier papers by these authors (HESS, 2016; WRR, 2017). This manuscript presents interesting (modelling) insights on the development of freshwater bodies below these beach nourishments. The content is scientifically relevant, of good quality, and properly presented. Some minor revisions are, however, required."

We would like to thank the Referee for the comments, which are highly appreciated.

[Figure]

We will improve the raised issues.

General Comments

"1. Being the third modelling paper on this topic in a row. The reader would be pleased with a clear picture of what is presented in which paper and what the differences between the applied groundwater models are. The WRR paper isn't even mentioned in the current manuscript."

Yes, we fully agree with the Referee and have added a brief description of the outcomes of the previous studies, and the connection with this third paper to the introduction.

"2. The reader would be pleased with more quantitative information, mainly in Section 3, especially on the freshwater volume developed. For instance: not 'substantially higher fresh groundwater volumes' (P11, line 11/12), but a percentage of volume increase. Another example: P10, line 20 and 21."

We agree with the Referee, and have provided more quantitative information in Section 3.

"3. Parts of Section 3 would better fit in Section 2."

We agree with the Referee, and have moved parts of section 3 to section 2. See responses to the specific comments.

Specific Comments

"- Title (and rest of manuscript): is 'resources' really the best term? I would associate that with use (abstraction) of the freshwater, which is not the case. Perhaps 'lenses' or 'bodies' are better alternatives"

With this term we wanted to both describe the lens/body of fresh groundwater and stress the potential for the growth of fresh groundwater resources (for abstraction) in mega-scale beach nourishments. However, we agree that 'lenses' fits better with the content of the ms and have changed 'fresh groundwater resources' to a 'fresh groundwater lens'.

"- Intro: first sentence can be removed"

We have removed the first sentence of the abstract.

"- P1, line 17: remove 'similarly,'"

We have replaced 'similarly' with 'whereas', to highlight the contrast with the previous line.

"- P2, line 2, replace 'clean' by 'high-quality'"

We have replaced 'clean' with 'high-quality'.

"- P2, line 14: do you really mitigate 'climate'? Suggest to replace 'climate' by 'climate change'."

Climate was added as a clarification: mitigating climate change. However, within the context of the paragraph we believe this is an unnecessary addition, and there 'climate' was removed.

"- P3, line 6. remove 'this study analyses' and add 'are analysed in this study' after 'fresh groundwater'"

In response to a comment of Anonymous Referee #2 we have deleted the entire paragraph (L5-12)

"- P3, line 7: remove e.g. and state only what you have analysed"

In response to a comment of Anonymous Referee #2 we have deleted the entire paragraph (L5-12)

"- P3, line 8: 'In this paper are described' instead of 'This paper describes'"

In response to a comment of Anonymous Referee #2 we have deleted the entire paragraph (L5-12)
"- P3, line 26: add 'groundwater' before 'model'"

We have added 'groundwater' before 'model' in line 26.

"- P3, line 29: add 'and salt' after 'fresh'"

We have added 'and salt after 'fresh in line 29.

"- P4, line 10: use 'Aquifer 1a' instead of 'aquifer 1a'. Some holds for 'Monitoring well'. Capitals"

In accordance with the Figures, we have change all references to numbered items (e.g. Aquifer 1a and Monitoring well 1) to capitals.

"- P4, line 24-28: Vague, please add a figure to support this section"

We have definitions of mentioned areas (e.g. foredunes) to Figure 2, and have added a dotted line to Figure 2 that signifies the initial salinity distribution (fresh/salt).

"- P4, 33: replace 'maintain' by 'keep'"

We have replaced 'maintain' with 'keep'.

"- P4, line 34: why is this control of groundwater direction needed?"

This control was needed to prevent a flow of previously infiltrated saline groundwater and possibly contaminated groundwater to the drinking water wells. This is briefly mentioned in the previous sentence as 'prevent any undesirable or unexpected effects of the construction of the Sand Engine to the groundwater quality (e.g. flow of saline groundwater towards pumping wells).

"- P6, line 16: where was the wave set-up and run-up based on?"

The simulation of the land-surface inundations (incl. wave set-up and run-up) was based on a method that was described and evaluated in a previous paper (Huizer et al., 2017). This method was based on the estimation of the wave set-up and run-up heights of a paper of Stockdon et al. (2006). We have added a statement and reference
to this paper to the start of section 2.5.1.

"- P6, line 18-22: make schematization and give max, min, average."

This method was described and evaluated more extensively in a previous paper (Huizer et al., 2017). We have added a statement and reference to this paper to the start of section 2.5.1.

"- P6, line 28: no water level measurements available in the lagoon?"

Yes, in the lagoon we only had water levels for the period of 2 to 23 October 2014 (see Figure 5).

"- P7, line 21: 'smaller-sized' –> quantify"

The model stress period varied between 10 minutes to multiple hours, dependent on observed changes in seawater level. For each stress period the average groundwater recharge was calculated with the hourly precipitation and evaporation. This is likely obvious for most readers, and therefore the line was removed from the ms.

"- P8, line 21: sentence missing?" No, the two dots were a result of a mistake in the handling of track changes. We removed one of the dots, and have rephrased the line after the dots, to create a better connection between the lines.

"- P9, line 8: did you check of these changes in abstraction rates might have occurred?"

No, we were unable to obtain a timeseries of extraction rates for the mentioned period.

"- P9, line 15-21: This part more or less just drops in here. Partly methods..."

We agree, and have moved the part that belongs in the method chapter (lines 15 – 18) to section 2.3, and have connected the remaining lines 18 – 21 to the next paragraph.

"- P9, line 28-29: Please provide more information on this initial groundwater salinity (figure)"

We have definitions of mentioned areas (e.g. foredunes) to Figure 2, and have added

a dotted line to Figure 2 that signifies the initial salinity distribution (fresh/salt).

"- P10, line 1-4: consider to move this to methods section"

We agree, and have moved these lines, together with lines 31-33 on P9, to the methods section.

"- P10, line 21-30: separate field observation and models results. Now, this part is confusing. Start with field observations."

We have modified and rephrased lines 21-30.

"- P11, line 21: A1 or A2? C1 or C2?"

We meant model scenario A2, C1, C2: the water level in het lagoon is less significant than wave set-up/run-up, and the extinction depth.

"- P11, line 29: replace 'will be' be 'was'"

We have added a line before line 29, and replaced 'will be' with 'was therefore'.

"- P12, line 29: quantify"

We have rephrased the line to: '... in the Sand Engine, at the monitoring wells the underestimations range between -0.4 and -0.6 m.'

"- P13, line 6: North Sea salinity: speculative, you don't know if this can really explain deviations"

We agree that we don't know if this can explain the observed deviations, and here the potential influence was probably small. Therefore, we have deleted this statement from the sentence.

"- P13, line 9: Why aren't there more measurements?"

We would have liked to get more salinity measurements, however during the research on the Sand Engine we prioritized other measurements (see previous paper in WRR).

"- P13, line 31: any reference to studies with comparable approach?"

We found a few studies (Falkland and Woodroffe, 2004 and Post and Houben, 2017) that used a comparable approach and added them in section 2.5.3.

"- P14, line 2: what is the basis for 'one to three monthly'"

The topographic surveys were executed every 1 to 3 months in the period of 2011-2016. We have rephrased the sentence to create more clarity.

"- P14, line 3: what is a 'unsaturated groundwater model'?"

We have replaced 'model' with 'flow'. The sentence refers to models that specifically include unsaturated groundwater flow (e.g. Richards equation).

"- P14, line 6: 'the study area' by 'a mega-scale beach nourishment (the Sand Motor)'"

We replaced 'the study area' with 'a mega-scale beach nourishment (the Sand Engine)'

"- P14, line 10: Please introduce these bullets with one sentence."

We have replaced the last dot with a colon to show the connection with the line above the bullets.

---

## Author Comment (AC2) · 9 Nov 2017

"This manuscript shows the application of numerical models to simulate different scenarios for the quantification of the freshwater resources in a mega beach nourishment experience in the Dutch coast. This experiment, also called the Sand engine, has different challenges for the simulation of the groundwater flow since it is a highly dynamic environment that affects to the shape and boundary conditions, this is an interesting topic in the hydrogeological sciences and many readers can be interested on seeing the results."

We would like to thank the Referee for the comments, which are highly appreciated.

[Figure]

We will improve the raised issues.

General Comments

"It is well written and applies the correct methods. But I think that the manuscript can be improved by showing the results with a more global perspective and less as a case study description. Also the novelty of this study should be presented more clearly and the analysis of the results/discussion can be improved."

Thanks for the compliments. We believe that the particular conditions at a coastal site are very important, which makes it difficult to extrapolate the effects to a global perspective. Therefore we wrote the paper more from the perspective of the Sand Engine, to describe the issues at this site in detail, and highlighted briefly the potential issues for other sites.

"The aims are very broad and the particular effects of coastal forcing and the geomorphological changes can be better described. I think that introducing the challenges from the perspective of the development of the numerical model would be needed to have a more complete overview. A more extensive introduction would help to understand the tasks that are going to be solved later.'

We agree, and have rephrased the aims, and introduces the challenges from the perspective of the development of a numerical model.

"In general it seems to be repetitive and a bit ambiguous the description of coastal forcing with multiple mentions of how important is this instead of describing precisely to what aspects is refereeing."

We agree with the Referee, and have deleted some sentences regarding coastal forcing.

"The boundary conditions of the model are based on previous models, unless this model is just a modification of a previous model (in that case it should be said), the boundary conditions should be clearly established. For example: "Other specified

heads and concentration boundaries were determined with an additional simulation with the groundwater model as described in Huizer (2016)" Which are the other specified heads and concentrations? What model scenarios are referring here? (the scenarios are presented later in the manuscript but at this point is not clear) ": : : the former model underestimated the salinization: : :" which model? In general this paragraph can be rewritten under the consideration that the reader does not have to be necessarily familiar with the previous models in the area. Probably a full description of previous models and the novelties of this study would help to the reader to frame better this study."

We have rewritten some sections of the paragraph to clarify the methodology.

"It is not clear what are the areas presented as foredunes, dunes, beach and sand engine, they should be defined and presented in the map."

We have added definitions of the mentioned areas (e.g. foredunes) to Figure 2, and have added a dotted line to Figure 2 that signifies the initial salinity distribution (fresh/salt).

"The model calibration presents some very clear criteria combined with ambiguous and arbitrary i.e. "the variation in the simulated head should be close or (almost) identical to the observed fluctuation pattern." What is close or identical? This is a very arbitrary description that can be improved. Another element that is arbitrary in the calibration is: "the salinity should be small or otherwise explicable". What is small? What is explicable?"

We agree that these criteria could be interpreted in various ways, therefore we have made small modifications in the phrasing of the criteria. The intent was the couple the mentioned criteria: the transient error in groundwater head should be smaller than the observed standard deviation, but stay similar to the observed fluctuations. We chose to keep the criteria concerning the groundwater salinity more qualitative, because the number of measurements was limited and small variations in the depth of the interface

can lead to large errors in the simulated groundwater salinity.

"It is defined that six factors affect to fresh groundwater resources but only three are considered. It is not clear what is the criteria for this and if this would also affect to the results obtained. I think that a better introduction would help to understand this."

We have rephrased this paragraph and deleted the line on the six factors. Together with the improvement to the introduction we believe this is resolved.

"The discussion is too descriptive basically presenting the results of the different models and adding some elements that could affect to the models (and in most cases are minor). A probably more interesting discussion, that can be also included in the conclusions, would be a quantitative comparison between the different factors that have been presented in this work. This would generate a broader impact of the results."

We agree that some aspects of the paper are not presented clear enough, therefore we have made some modifications to the discussion. Together with the improved descriptions of the aim of the research in the introduction, we believe this improved the impact of the results. However, we chose to avoid a too quantitative approach in the discussion, because some of the differences between the factors could be related to the conditions at this particular site, and some differences between scenarios could be caused by the differences in nature of the scenarios (how to compare a constant sea-level, to a constant morphology).

Specific Comments

"Page 3. Lines 5-12. The description about the outline of the work is not needed."

Yes, we agree and have deleted these lines.

"Page 6. Lines 20-22. Which data were used for this calculation?"

We used offshore measurements at the "Euro platform" of the Ministry of Infrastructure and the Environment (Rijkswaterstaat). This was added to the lines.
"Page 8. Line 21. There are two dots in a row."

We have corrected this in the ms.

"Page 9. Line 32. Either mention as described in sect 2.2., or repeat the model discretization but both are repetitive."

We agree, and have moved these lines, together with lines 31-33 on P9, to the methods section.